# Poly-Visual-Expert Vision-Language Models

**Xiaoran Fan**[*], **Tao Ji**[*], **Changhao Jiang**[*], **Shuo Li**[*], **Senjie Jin**[*], **Sirui Song**
**Junke Wang, Boyang Hong, Lu Chen, Guodong Zheng, Ming Zhang, Caishuang Huang**
**Rui Zheng, Zhiheng Xi, Yuhao Zhou, Shihan Dou, Junjie Ye, Hang Yan**
**Tao Gui**[†]**, Qi Zhang,**[†]**Xipeng Qiu, Xuanjing Huang, Zuxuan Wu, Yu-Gang Jiang**
Fudan NLP Lab & Fudan Vision and Learning Lab
Shanghai 200438, China
`xrfan23@m.fudan.edu.cn,{tgui, qz}@fudan.edu.cn`

## Abstract

Current large vision-language models (VLMs) frequently face challenges such as the limited capabilities of a single visual component and the excessive length of visual tokens. These issues can limit the model's ability to interpret complex visual information and over-lengthy contextual information accurately. Tackling these challenges is crucial for enhancing the performance and applicability of VLMs. This paper proposes leveraging the ensemble experts technique to synergize the capabilities of individual visual encoders, including those skilled in image-text matching, image segmentation, OCR, etc. This method introduces a fusion network that consolidates the outputs from different visual experts while bridging the gap between image encoders and pre-trained LLMs. In addition, we explore different positional encoding schemes to mitigate the waste of positional encoding caused by lengthy image feature sequences, effectively addressing the issue of position overflow and length limitations. For instance, in our implementation, this technique significantly reduces the positional occupancy in models like SAM, from a substantial 4096 to a more efficient 64 or even down to 1. Experimental results show that VLMs with multiple experts consistently outperform isolated visual encoders, with notable performance improvements as more experts are integrated. Our codes are available on our project website.[1]

## 1  Introduction

Current large vision-language models (VLMs) demonstrate significant potential in tasks requiring joint visual and linguistic perception, such as image captioning (Agrawal et al., 2019), visual question answering (Antol et al., 2015), visual grounding (Yu et al., 2016), and autonomous agents (Durante et al., 2024; Xi et al., 2023). VLMs harness large language models (LLMs) as cognitive foundation models to empower various vision-related tasks, while **one vision component**, such as CLIP (Radford et al., 2021), typically serves as auxiliary modules that provide additional visual perception (Liu et al., 2023b). However, the perception abilities of the individual vision models still lag behind, even in simple tasks like counting. (Yamada et al., 2022; Thrush et al., 2022; Yuksekgonul et al., 2022). This gap highlights a significant limitation in these models' capacity to process and understand visual information as effectively as they handle linguistic data. According to the operation of the vertebrate visual system, with each functional unit encoding different visual aspects in parallel, retinal ganglion cells transmit distinct features to the brain (Baden et al., 2016). This biological mechanism suggests **a model structure where the varied visual information should be parallelly encoded by diverse perception channels.**

---

[*] Equal contributions.
[†] Corresponding author.
[1] `https://github.com/FudanNLPLAB/MouSi`

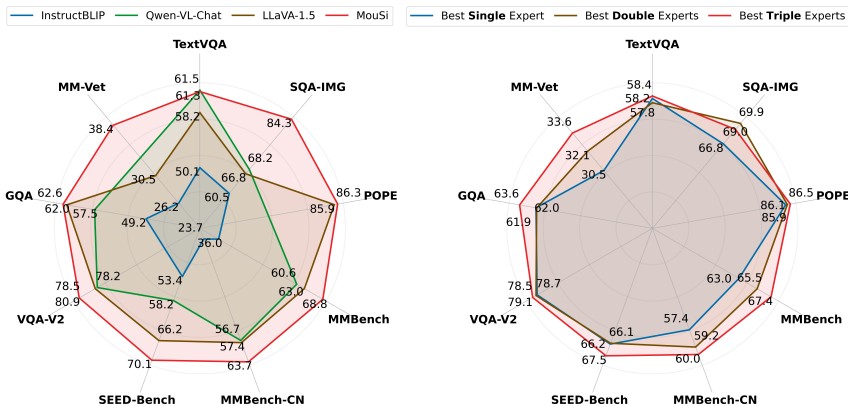

Figure 1: **Left**: Comparing InstructBLIP, Qwen-VL-Chat, and LLaVA-1.5-7B, our poly-visual-expert achieves SoTA on a broad range of nine benchmarks. **Right**: Performances of the best models with different numbers of experts on nine benchmark datasets. Overall, triple experts are better than double experts, which are better than a single expert.

To this end, the community has verified that each model, with its unique vision processing approach contributes differently to understanding visual content (Chen et al., 2023a). CLIP, with its contrastive learning approach, excels in aligning images with textual descriptions, providing a robust semantic understanding (Radford et al., 2021). DINO[v2], through its self-supervised learning at both the image level and patch level, offers significant advances in robust and stabilized feature extraction without relying on labeled data (Oquab et al., 2023). LayoutLM[v3]'s specialization in document AI tasks demonstrates the power of visual text processing (Huang et al., 2022). (Wang et al., 2023a) empirically investigated different visual tokenizers pre-trained with dominant methods (i.e., DeiT (Touvron et al., 2021), CLIP, MAE (He et al., 2021), DINO (Caron et al., 2021)), and observed that CLIP could capture more semantics, whereas the other models excelled at fine-grained perception. However, on the multimodal leaderboard organized by OpenCompass[2], the visual encoders of all open-source VLMs are based on the pre-trained CLIP encoder family. Many researchers have pointed out the shortcomings of the CLIP encoder, such as the inability to reliably capture even basic spatial factors of images (Kamath et al., 2023), suffering from object hallucination (Li et al., 2023c), and so on. In light of the distinct capabilities and limitations of these diverse vision models, a key question emerges: **How can we incorporate the strengths of multiple visual experts so that they work in synergy to improve overall performance?**

Drawing inspiration from biology, we take on the poly-visual-expert perspective and design a novel model, similar to how the vertebrate visual system operates. Consequently, in the process of developing VLMs with poly-visual experts, three problems are in major concern: (1) whether the poly-visual experts are effective; (2) how to better integrate multiple experts; and (3) how to avoid exceeding the LLM's maximum length with multiple visual experts?

In order to verify whether multiple visual experts are effective for VLMs, we construct a candidate pool consisting of six well-known experts, including CLIP, DINO[v2], LayoutLM[v3], Convnext (Woo et al., 2023), SAM, and MAE. Using LLaVA-1.5 as the base setup, we explored single-expert, double-expert combinations, and triple-expert combinations in nine benchmarks. The results, as shown in Figure 1, indicate that as the number of visual experts increases, the VLMs acquire richer visual information (due to more visual channels), and the upper limit of the multimodal capability improves across the board.

In existing single visual channel VLMs, visual signals are transmitted using either the MLP network (Liu et al., 2023a; Wang et al., 2023b) or the Q-Former network (Bai et al., 2023; Dai et al., 2023b). To support multi-channel signal transmission from multiple experts, we modified both methods for poly-expert fusion networks separately. The proposed method also

---

[2]https://rank.opencompass.org.cn/home

compresses the local visual information by multi-patch-one-token for better transmission efficiency and reduces the quadratic computational cost of subsequent processing of VLMs.

In position-aware VLMs, vision tokens consume a staggering amount of positional embeddings. Taking a single-turn multimodal dialogue in VQA as an example, with the SAM expert, the number of vision tokens (about 4096) is more than 500 times higher than the number of text tokens (about 8.7). Inspired by the fact that visual experts already have positional encodings, we believe it is redundant to again assign a VLM position embedding to each visual token individually. Therefore, we explore different positional encoding schemes to effectively address the issue of position encoding waste. The results show that the two schemes: sharing one position for all patches and 2D positional encoding (rows plus columns) are able to reduce the position consumption (in the case of CLIP, the PE used drops from 576 to 24 or even 1), while the performance is still comparable.

Our contributions can be summarized as follows:

- We introduce a poly-visual-expert VLM that synergistically combines the strengths of various visual encoders to improve the overall capabilities of VLMs.
- We tackle the challenge of vision token overflow in VLMs by proposing multi-patch-single-token projection and efficient positional encoding solutions.
- By experimenting with different combinations of experts, our results demonstrate enhanced performance (+1.8 with fair comparison) in multimodal tasks.

## 2 Architecture

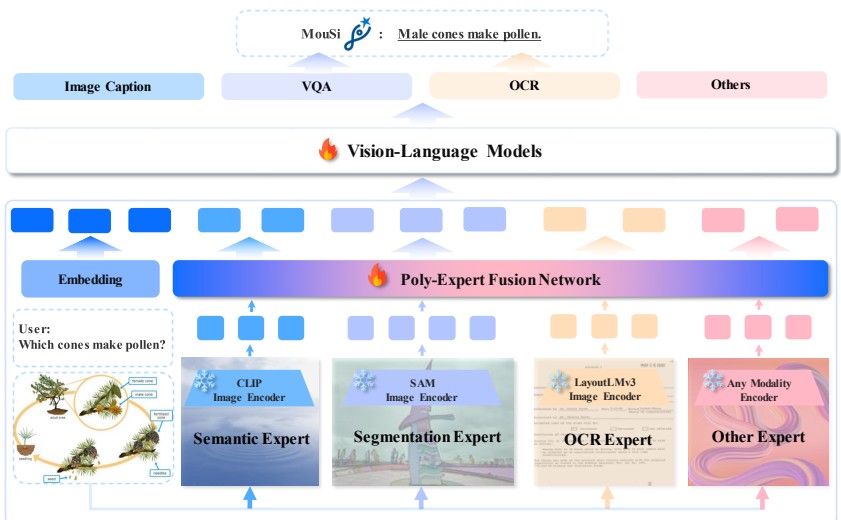

Figure 2: An overview of our model structure. The poly-vision-expert model supports the integration of visual experts with various types and capabilities.

### 2.1 The Overview

When a user uploads an image of wind pollination in a conical inflorescence and asks "Which cones make pollen?" the image is processed in sequence through the encodings of the CLIP expert, the SAM expert, and the LayoutLM expert, yielding three sets of visual representations. Subsequently, a poly-expert fusion network compresses the multi-channel visual information and aligns it multimodally to the vision input tokens for our poly-vision-expert model. The user's question is processed into text tokens by the LLMs' Embedding layer. Finally, our model generates the correct answer "Male cones make pollen." by employing its VQA capabilities to understand the vision-language question, and its OCR capabilities to recognize the answer text from the image.

| Expert | Res. | Param. | d_hid | #Patch | Type | Pre-training Tasks | Pre-training Images |
|---|---|---|---|---|---|---|---|
| CLIP | 336 | 300M | 1024 | 576 | ViT | Image-Text Matching | 400M |
| DINOv2 | 224 | 1.1B | 1536 | 256 | ViT | DINO+iBOT+SwAV | 142M |
| LayoutLMv3 | 224 | 368M | 1024 | 196 | ViT | Document OCR | 11M |
| ConvNeXt | 384 | 200M | 768 | 1024 | CNN | Image Classification | 2B |
| SAM | 1024 | 637M | 1280 | 4096 | ViT | Image Segmentation | 11M |
| MAE | 224 | 630M | 1280 | 256 | ViT | Patch-level Denoising | 1.3M |

Table 1: Comparison of six pre-trained visual experts. **Res.** indicates image resolution, **d_hid** indicates hidden dimension and **Param.** indicates the number of parameters.

In order to accomplish the above task, we propose the poly-vision-expert model, which consists of three fundamental components:

1. a multi-expert visual encoder, which combines the experts selected from a pool;
2. a poly-expert fusion network, which is implemented as a simple projection fusion method or a Q-Former fusion method (Li et al., 2023b);
3. a pre-trained open-source LLM (e.g., *Vicuna v1.5*).

Figure 2 shows an overview of our poly-vision-expert model. The core of a Vision-Language Model is typically an LLM which is pre-trained on large-scale textual corpus. To perceive the visual signals, a vision encoder and vision-language connection layer are adopted to separately extract the visual features and align them to the semantic space of LLM.

The VLM takes as input a sequence comprised of interleaved text and image segments, denoted as $X = (\ldots, T_1, I_1, T_2, I_2, \ldots)$, where text fragments $T$ are processed by the tokenizer and embedding layer of the LLM, and image segments $I$ are fed to the vision encoder. To ensure the universality and generalizability of the vision encoder, it is common practice to freeze its pre-trained parameters. In this paper, we rethink the design of the visual encoder in VLMs and aim to improve its capability by ensembled experts.

## 2.2  Multi-Expert Vision Encoder

After extensive investigation, we choose six vision encoders skilled in different domains, including CLIP (2021), DINO$^{v2}$ (2023), LayoutLM$^{v3}$ (2022), Convnext (2023), SAM (2023), and MAE (2021). They differ significantly from each other in terms of input resolution, hidden size, model type, model size, pre-training tasks, and training methods, as shown in Table 1. Their detailed descriptions can be found in Appendix A.

Given a image $I$ in the input sequence and a vision expert encoder $e_i(\cdot)$, we can obtain the representation vectors of $n$ image patches:

$$v_1^i, v_2^i, \ldots, v_n^i = e_i(I). \tag{1}$$

Assuming we have three experts ($e_i(\cdot) \in \mathbb{R}^{n_i \times d_i}$, $e_j(\cdot) \in \mathbb{R}^{n_j \times d_j}$, $e_k(\cdot) \in \mathbb{R}^{n_k \times d_k}$), the final sequence of image representations $V_I$ is a concatenation of the three output sequences.

$$V_I = e_i(I) \oplus e_j(I) \oplus e_k(I) = [v_1^i, \ldots, v_{n_i}^i, v_1^j, \ldots, v_{n_j}^j, v_1^k, \ldots, v_{n_k}^k] \tag{2}$$

It is worth noting that each expert outputs a different number ($n_i$ vs. $n_j$ vs. $n_k$) and dimension ($d_i$ vs. $d_j$ vs. $d_k$) of vectors, and we will handle these differences in the fusion network.

## 2.3  Poly-Expert Fusion Network

Since the dimension and number of output sequences are often different for different visual experts, a fusion network needs to be designed to unify the processing. Following LLaVA (Liu et al., 2023b) and BLIP (Li et al., 2022), we propose an MLP projection fusion network and a Q-Former fusion network, respectively.

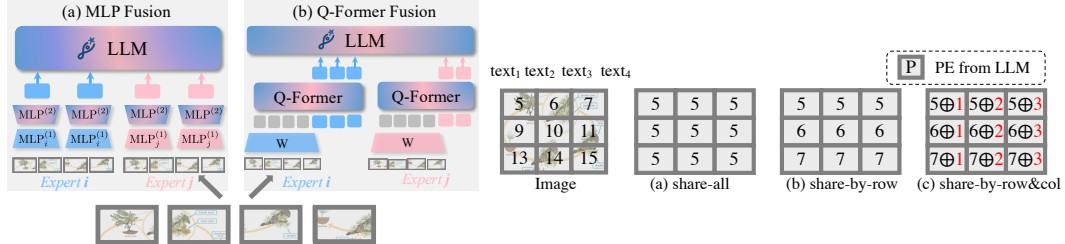

Figure 3: **Left:** Examples of two poly-expert fusion methods. We show how the MLP method compresses visual information with "2-patches-1-token", and how the Q-Former method compresses information with 3 trainable queries. The modules with color gradients represent the sharing of parameters among multiple experts to transfer knowledge. **Right:** Diagram of the four positional encoding schemes. The $\oplus$ operator indicates that the row position embedding and column position embedding are summed.

**MLP projection** is a 2-layer ($d_{in} \to d_{hidden} \to d_{out}$) multilayer perceptron. To simplify the processing and to share the knowledge among multiple experts, we set the hidden dimension ($d_{hidden}$) and the output dimension ($d_{out}$) equal to the dimension ($d_{model}$) of the LLM, and the second layer network (MLP$^{(2)}$ : $d_{hidden} \to d_{out}$) parameters are shared among all experts. Given a specific expert $e_i(\cdot)$, the first layer network is defined as MLP$_i^{(1)}$ : $d_i \to d_{hidden}$.

$$H_I = \text{MLP}_i^{(1)}\left(e_i(I)\right) \oplus \text{MLP}_j^{(1)}\left(e_j(I)\right) \oplus \text{MLP}_k^{(1)}\left(e_k(I)\right)$$
$$V_I = \text{MLP}^{(2)}(H_I) \tag{3}$$

In practice, multiple experts output a large number of vision tokens, which not only increases the computational cost and memory usage of the VLM but also tends to exceed the maximum length limit during inference. Therefore, we propose **multi-patches-one-token** projection to proportionally reduce the number of tokens output by each expert. Since image signals have local or sparse properties, it is practical to use one token to represent neighboring patches. Take $m$-patch-one-token for example, we make the input dimension of the first layer of the network m times (MLP$^{(1)}$ : $d_{in} \times m \to d_{hidden}$), and its hidden layer output vectors $h_1^i, h_2^i, \ldots$ are defined as follows:

$$h_1^i = \text{MLP}^{(1)}\left(\left[v_1^i \oplus \cdots \oplus v_m^i\right]\right), \; h_2^i = \text{MLP}^{(1)}\left(v_{m+1}^i \oplus \cdots \oplus v_{2m}^i\right), \ldots$$

where the $[\oplus \cdots \oplus]$ notation denotes concatenation over the vector dimension. The final number of vision tokens is reduced to $\frac{1}{m}$ of the original. In practice, $m$ is typically set from 2 to 8, which reduces cost while usually not losing performance on downstream tasks. If $m$ is set too large, the information of the image might be lost.

**Q-Former network** is a trainable Querying Transformer module and proposed to bridge the gap between a frozen image encoder and a pre-trained LLM. It extracts a fixed number of output features from the vision encoder, independent of input image resolution. We create a set number of learnable query embeddings as input to the Q-Former. The queries interact with each other through self-attention layers, and interact with frozen image features $e_i(I)$ through cross-attention layers. The output queries of the last layer are projected to the input layer of the LLM. We use the pre-trained parameters in BLIP-2 as initialization to accelerate convergence and, similar to the second layer MLP network, share the parameters among all experts. Since the dimension of query embeddings is equal to 768, we add an additional linear transformation ($W_i \in \mathbb{R}^{d_i \times 768}$) for each expert.

$$V_I = \text{Q-Former}\left(W_i\left(e_i(I)\right) \oplus W_j\left(e_j(I)\right) \oplus W_k\left(e_k(I)\right)\right)$$

The ablation study in Section 3.2.1 shows that the MLP fusion network fuses better than the Q-Former despite having fewer parameters and not being pre-trained.

## 2.4 Different Positional Encoding Schemes

Inspired by the fact that visual experts already have positional encodings (e.g., 2D position encoding in ViT (Wang & Liu, 2019)), we believe it is redundant to assign a VLM position embedding (PE) again to each visual token individually. Figure 3 illustrates our exploration of three encoding schemes to optimize PE assignments:

1. all vision tokens of an image share a PE (*share-all*);
2. one PE shared by the same row of vision tokens (*share-by-row*);
3. one PE shared by the same row of vision tokens, plus a set of learnable columns PEs (*share-by-row&col*).

Among the three methods, *share-all* can reduce the original $O(N^2)$ PE cost to $O(1)$, while the *share-by-row* and *share-by-row&col* can reduce the cost to $O(N)$. All of them can significantly alleviate the out-of-maximum-length problem, but the question is **how much do they affect the performance of VLM?** We report ablation results in Section 3.2.2.

# 3 Experiments

## 3.1 Main Results

We conduct explorations of single-expert, double-expert, and triple-expert ensembles. Following LLaVA-1.5 (Liu et al., 2023a), our training pipeline consists of two stages. In the pre-training stage, we freeze the text-only LLM and the multi-expert encoder, and train the poly-visual fusion network from scratch to align the representation space of both. After training on a large-scale weakly-supervised (with noise) dataset, the text-only LLM is already capable of multimodal input and comprehension. In the fine-tuning stage, we unfreeze the LLM and further train it together with the poly-visual fusion network on diverse and high-quality supervised fine-tuning (SFT) datasets.

**Datasets & Evaluation & Hyperparameters**   We use the *same* datasets, evaluation, and hyperparameters as in LLaVA-1.5 (Liu et al., 2023a). The details can be found in Appendix C.

**Case Study**   We present case studies on seven tasks in Appendix D. Our model can successfully follow multimodal instructions, enabling flexible human interaction.

### 3.1.1 Single Vision Expert

| Model | Param | VQA$^{v2}$ | GQA | SQA$^I$ | VQA$^T$ | POPE | MMB | MMB$^{CN}$ | SEED$^I$ | MM$^{Vet}$ | **Avg** |
|---|---|---|---|---|---|---|---|---|---|---|---|
| | | | | | *Single Expert* | | | | | | |
| CLIP | 7.3B | **78.5** | **62.0** | **66.8** | **58.2** | **85.9** | **63.0** | **57.4** | **66.2** | **30.5** | **63.2** |
| DINO$^{v2}$ | 8.1B | 74.9 | 61.7 | 66.1 | 46.2 | 84.6 | 57.9 | 48.7 | 63.4 | 23.4 | 58.5 |
| LayoutLM$^{v3}$ | 7.4B | 44.9 | 40.0 | 62.8 | 43.6 | 59.1 | 29.0 | 19.8 | 34.8 | 11.8 | 38.4 |
| ConvNeXt | 7.2B | 75.1 | 60.5 | 65.0 | 56.3 | 85.6 | 63.3 | 55.0 | 61.5 | 26.0 | 60.9 |
| SAM | 7.6B | 64.7 | 55.8 | 63.9 | 44.1 | 82.0 | 43.7 | 33.9 | 51.9 | 17.7 | 50.9 |
| MAE | 7.6B | 62.0 | 53.2 | 63.3 | 44.5 | 79.7 | 41.6 | 33.0 | 49.4 | 16.5 | 49.2 |

Table 2: **Comparison of six vision experts on 9 benchmarks.**

Table 2 compares the performance of all six VLMs with a single vision expert. The CLIP expert achieves the best performance in 8/9 benchmarks, fully explaining why it has become the dominant choice of vision encoder for VLMs. Comparing the different attributes of the experts, CLIP ranked 5th in terms of the number of parameters, 3rd in terms of image resolution, and 2nd on the size of the pre-training data, none of which had an absolute lead. Therefore, we guess that its main advantage lies in its image-text matching pre-training task, which has multimodal alignment capability in advance. Overall, the performance ranking of the six experts is roughly CLIP>ConvNeXt>DINO$^{v2}$>SAM>MAE>LayoutLM$^{v3}$. In

addition, LayoutLM$^{v3}$ is an undisputed expert in OCR and SAM in image segmentation but performs poorly as a single visual encoder in VLM. A natural question is *whether poly-expert fusion can activate their capabilities in their specialized fields?*

### 3.1.2 Double Vision Experts

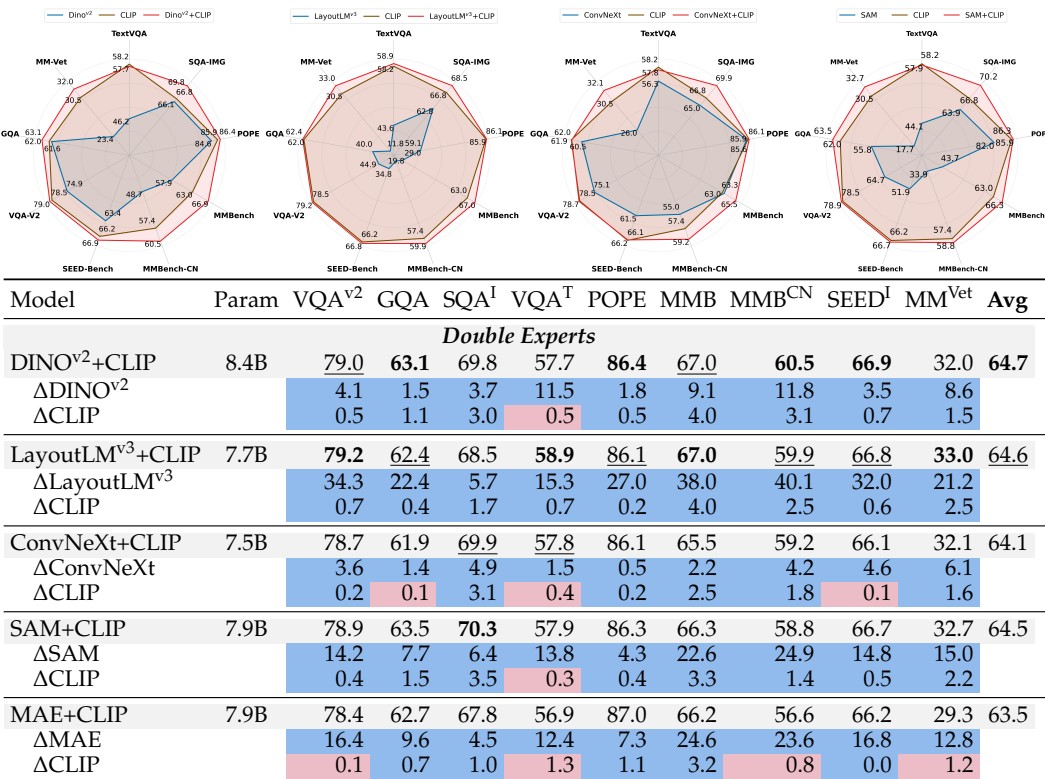

| Model | Param | VQA$^{v2}$ | GQA | SQA$^I$ | VQA$^T$ | POPE | MMB | MMB$^{CN}$ | SEED$^I$ | MM$^{Vet}$ | Avg |
|---|---|---|---|---|---|---|---|---|---|---|---|
| | | | | | *Double Experts* | | | | | | |
| DINO$^{v2}$+CLIP | 8.4B | 79.0 | 63.1 | 69.8 | 57.7 | 86.4 | 67.0 | 60.5 | 66.9 | 32.0 | 64.7 |
| ΔDINO$^{v2}$ | | 4.1 | 1.5 | 3.7 | 11.5 | 1.8 | 9.1 | 11.8 | 3.5 | 8.6 | |
| ΔCLIP | | 0.5 | 1.1 | 3.0 | 0.5 | 0.5 | 4.0 | 3.1 | 0.7 | 1.5 | |
| LayoutLM$^{v3}$+CLIP | 7.7B | 79.2 | 62.4 | 68.5 | 58.9 | 86.1 | 67.0 | 59.9 | 66.8 | 33.0 | 64.6 |
| ΔLayoutLM$^{v3}$ | | 34.3 | 22.4 | 5.7 | 15.3 | 27.0 | 38.0 | 40.1 | 32.0 | 21.2 | |
| ΔCLIP | | 0.7 | 0.4 | 1.7 | 0.7 | 0.2 | 4.0 | 2.5 | 0.6 | 2.5 | |
| ConvNeXt+CLIP | 7.5B | 78.7 | 61.9 | 69.9 | 57.8 | 86.1 | 65.5 | 59.2 | 66.1 | 32.1 | 64.1 |
| ΔConvNeXt | | 3.6 | 1.4 | 4.9 | 1.5 | 0.5 | 2.2 | 4.2 | 4.6 | 6.1 | |
| ΔCLIP | | 0.2 | 0.1 | 3.1 | 0.4 | 0.2 | 2.5 | 1.8 | 0.1 | 1.6 | |
| SAM+CLIP | 7.9B | 78.9 | 63.5 | 70.3 | 57.9 | 86.3 | 66.3 | 58.8 | 66.7 | 32.7 | 64.5 |
| ΔSAM | | 14.2 | 7.7 | 6.4 | 13.8 | 4.3 | 22.6 | 24.9 | 14.8 | 15.0 | |
| ΔCLIP | | 0.4 | 1.5 | 3.5 | 0.3 | 0.4 | 3.3 | 1.4 | 0.5 | 2.2 | |
| MAE+CLIP | 7.9B | 78.4 | 62.7 | 67.8 | 56.9 | 87.0 | 66.2 | 56.6 | 66.2 | 29.3 | 63.5 |
| ΔMAE | | 16.4 | 9.6 | 4.5 | 12.4 | 7.3 | 24.6 | 23.6 | 16.8 | 12.8 | |
| ΔCLIP | | 0.1 | 0.7 | 1.0 | 1.3 | 1.1 | 3.2 | 0.8 | 0.0 | 1.2 | |

Table 3: Comparison of different **double-expert** encoders. The Δ-marked rows are compared to the single-expert methods. Where blue cells mean the **double-expert** model is better, and red cells mean the **single-expert** model is better.

The current mainstream open-source VLMs have only one visual encoder, i.e., a single visual channel. Among six evaluated vision experts, the CLIP expert achieves the best performance in 8/9 benchmarks. However, multimodal tasks are diverse, and different tasks require different visual signals. We investigate whether dual-channel, i.e., double visual experts can outperform single experts on various tasks. We combine the strongest CLIP expert with other experts to construct a total of five double-expert encoders.

Table 3 shows the performance of the double-expert encoders and their respective single expert. The results show that the "DINO$^{v2}$+CLIP", "LayoutLM$^{v3}$+CLIP", and "ConvNeXt+CLIP" three double-expert encoders outperform the arbitrary single encoder in almost all cases (23/27). This demonstrates that using two visual channels enhances multimodal capabilities, confirming the feasibility of multi-expert collaboration.

Comparing the performance between double-expert methods, we found that the best double-expert is DINO$^{v2}$+CLIP, rather than the ensemble of the best single expert CLIP and the second-best ConvNeXt. It indicates that superior performance as a single expert does not necessarily imply optimality when ensembled. Since ConvNeXt and CLIP have considerable overlap in their training methods and training corpora, leading to the extraction of similar visual information, whereas the self-supervised DINO$^{v2}$ and the weakly-supervised CLIP complement each other, resulting in a more effective ensemble. Furthermore, it is worth mentioning that LayoutLM$^{v3}$, which performed the worst as a single expert, shows significant

| Model | Param | VQA$^{v2}$ | GQA | SQA$^I$ | VQA$^T$ | POPE | MMB | MMB$^{CN}$ | SEED$^I$ | MM$^{Vet}$ | **Avg** |
|---|---|---|---|---|---|---|---|---|---|---|---|
| | | | | *Triple Experts* | | | | | | | |
| ConvNeXt+LayoutLM$^{v3}$+CLIP | 7.9B | 78.5 | 63.3 | **70.2** | 58.0 | **87.3** | 66.8 | 58.9 | 66.0 | 32.2 | 64.6 |
| ΔConvNeXt+CLIP | | 0.2 | 1.4 | 0.3 | 0.2 | 1.2 | 1.3 | 0.3 | 0.1 | 0.1 | |
| ΔLayoutLM$^{v3}$+CLIP | | 0.7 | 0.9 | 0.9 | 1.7 | 1.2 | 0.2 | 1.0 | 0.8 | 0.8 | |
| ConvNeXt+DINO$^{v2}$+CLIP | 8.6B | 78.6 | 63.2 | 69.2 | 57.8 | 86.5 | 66.6 | 58.9 | 67.1 | 32.9 | 64.5 |
| ΔConvNeXt+CLIP | | 0.1 | 1.3 | 0.7 | 0.0 | 0.4 | 1.1 | 0.3 | 1.0 | 0.8 | |
| ΔDINO$^{v2}$+CLIP | | 0.4 | 0.1 | 0.6 | 0.1 | 0.1 | 0.4 | 1.6 | 0.2 | 0.9 | |
| LayoutLM$^{v3}$+DINO$^{v2}$+CLIP | 8.8B | **79.1** | **63.6** | 69.0 | **58.4** | 86.5 | **67.4** | **60.0** | **67.5** | **33.6** | **65.0** |
| ΔLayoutLM$^{v3}$+CLIP | | 0.1 | 1.2 | 0.5 | 0.5 | 0.4 | 0.4 | 0.1 | 0.7 | 0.6 | |
| ΔDINO$^{v2}$+CLIP | | 0.1 | 0.5 | 0.8 | 0.7 | 0.1 | 0.4 | 0.5 | 0.6 | 1.6 | |

Table 4: Performance comparison of different triple-expert methods. The Δ-marked rows are compared to the double-expert method. Where blue cells indicate the **triple-expert** model is better, and red cells indicate the **double-expert** model is better.

improvement when collaborating with CLIP, performing the best on four benchmarks and ranking overall just behind DINO$^{v2}$+CLIP. For SAM or MAE, even if their pre-training tasks are Image Segmentation, Patch-level Denoising, respectively, which are mainly optimization objectives from the image perspective. "SAM+CLIP" and "MAE+CLIP" double-expert encoders outperform the arbitrary single-encoder in almost all cases (13/18). Therefore, we can conclude that when paired with the versatile visual expert CLIP, other experts can focus on capturing supplemental visual information to further enhance performance.

### 3.1.3 Triple Vision Experts and More Vision Experts

As shown in Table 4, the triple-expert approach wins in 4/6 cases in comparison with the two-expert at the data size of LLaVA-1.5. The best-performing three-expert is LayoutLM$^{v3}$+DINO$^{v2}$+CLIP, followed by ConvNeXt+LayoutLM$^{v3}$+CLIP, and finally ConvNeXt+DINO$^{v2}$+CLIP. Among them, model LayoutLM$^{v3}$+DINO$^{v2}$+CLIP has the largest number of parameters, reaching 8.8 billion. Based on the triple-expert experiments, we further conceptualize scenarios with more experts. The different types of vision encoders we chose for the pre-training tasks contain Image-Text Matching, Image Classification, and Patch-level Denoising, which fall into three categories: weakly-supervised, self-supervised, and supervised. We have already achieved high performance in training triple-expert, and continuing to increase the number of experts has limited performance improvement. Additionally, the integration of extra experts leads to an excessive total token count, causing the length of the training process to be too long, increasing the cost of training and inferencing.

### 3.2 Ablation Study

#### 3.2.1 Effect of Fusion Methods

| Model | Param | VQA$^{v2}$ | GQA | SQA$^I$ | VQA$^T$ | POPE | MMB | MMB$^{CN}$ | SEED$^I$ | MM$^{Vet}$ | **Avg** |
|---|---|---|---|---|---|---|---|---|---|---|---|
| DINO$^{v2}$+CLIP+**MLP** | 8.4B | **79.0** | **63.1** | **69.8** | **57.7** | **86.4** | **67.0** | **60.5** | **66.9** | **32.0** | **64.7** |
| DINO$^{v2}$+CLIP+**Q-Former** | 8.5B | 60.4 | 50.9 | 66.7 | 45.1 | 45.2 | 52.7 | 44.8 | 51.8 | 20.5 | 48.7 |
| ConvNeXt+CLIP+**MLP** | 7.5B | **78.7** | **61.9** | **69.9** | **57.8** | **86.1** | **65.5** | **59.2** | **66.1** | **32.1** | **64.1** |
| ConvNeXt+CLIP+**Q-Former** | 7.6B | 65.8 | 52.6 | 68.7 | 45.6 | 77.0 | 59.7 | 49.8 | 53.2 | 22.1 | 54.9 |

Table 5: Performance comparison of different poly-expert fusion methods.

The MLP projection and Q-Former network are two mainstream methods for connecting vision and language. *Which of them can more effectively convey visual signals* is a key issue, especially in the context of poly-expert fusion. Table 5 presents the performance of using MLP and Q-Former respectively on three double-expert encoders, including "DINO$^{v2}$ & CLIP" and "ConvNeXt & CLIP". The results demonstrate that MLP significantly outper-

forms Q-Former in **all** cases, despite having fewer parameters and not utilizing pre-trained parameters like Q-Former, being instead directly initialized randomly.

### 3.2.2 Effect of Different Position Schemes

| Model | $VQA^{v2}$ | GQA | $SQA^I$ | $VQA^T$ | POPE | MMB | $MMB^{CN}$ | $SEED^I$ | $MM^{Vet}$ | Avg |
|---|---|---|---|---|---|---|---|---|---|---|
| Origin | 78.5 | 62.0 | 66.8 | 58.2 | 85.9 | 64.3 | 58.3 | 66.2 | 30.5 | 63.4 |
| Share-all | 79.0 | 62.4 | **68.4** | **58.4** | 86.3 | **67.4** | 58.2 | 65.7 | 31.7 | **64.2** |
| Share-by-row | 75.0 | 57.2 | 63.4 | 51.7 | 86.1 | 46.4 | 43.4 | 55.6 | **31.9** | 56.7 |
| Share-by-row&col | **79.0** | **62.6** | 68.3 | 58.1 | **86.3** | 66.0 | **58.8** | **66.2** | 30.6 | 64.0 |

Table 6: Comparison of four positional encoding schemes on 9 benchmarks.

Table 6 shows the results of four positional encoding schemes introduced in Section 2.4. The share-all method (used by CogVLM (Wang et al., 2023b)) not only saves the most PE but also improves the average performance by 0.8 on top of CLIP. The 2D positional coding (share-row&col) also improves the average performance by 0.6. However, share-row impairs the performance of the model, probably because row-sharing corrupts the position information of the visual encoder itself. The results support our conjecture that it is redundant to re-assign LLM positional encoding to each vision token that already has positional information.

### 3.3 Analysis

Among poly-visual encoders, an important question is the contribution of different experts to the model's output. Attention mechanisms are commonly used interpretive tools in Transformer networks (Wiegreffe & Pinter, 2019). Here, we employ a triple-expert encoder to assess the individual contributions of each expert within two multilingual benchmarks: MMB-English and MMB-Chinese. A sample's contribution is measured by the average attention the output token directs toward each expert's representations. By aggregating these averages across the dataset, we determine each expert's overall impact on the encoding process. Table 7 shows the individual contributions of the text prompt, $LayoutLM^{v3}$, $DINO^{v2}$, and CLIP to the output. The results indicate that the contribution of the text prompt to the answer is significantly higher than that of the visual experts. This is as expected. Firstly, the text prompt defines the format of the VLM's response, necessitating attention to the prompt during output, and secondly, the text has a higher information density than images, hence the average attention is usually higher for text. Comparing the three visual experts, we find that their contributions in descending order are CLIP, $DINO^{v2}$, and $LayoutLM^{v3}$. CLIP still demonstrates the characteristics of being the dominant eye or the primary visual channel. $DINO^{v2}$'s contribution is approximately 20% of CLIP's, while LayoutLM's contribution is minimal, at only 1% of CLIP's.

An ensuing inquiry is the relevance of visual channels that exhibit minimal contributions within the model's architecture. The detailed description of this section is in Appendix B.

### 3.4 VLM Benchmark results

We selected 9 of the 12 evaluation benchmarks for LLaVA-1.5[4]. Table 8 reports the results for LLaVA-1.5 (i.e., single CLIP expert), and our poly-vision-expert model ($LayoutLM^{v3}$+ConvNeXt+CLIP / $LayoutLM^{v3}$+$DINO^{v2}$+CLIP) on above benchmarks. The poly-vision-expert VLMs show improved performance over the single-expert VLM, with average improvements of +1.3/+1.8, while the number of parameters only increased by 600M/1500M. In comparison with established VLMs, our poly-vision-expert model emerges as the top-performing system in 7/8 benchmarks, and secures a close second in the remaining one, thus affirming its robust multimodal assistant capabilities.

---

[4]Excluding LLaVA-Bench that rely on unstable GPT4 responses, as well as VisWiz (2018) and MME (2023) for the website crashed.

|       | Prompt | LayoutLM$^{v3}$ | DINO$^{v2}$ | CLIP  |
|-------|--------|-----------------|-------------|-------|
| MMB   | 61.1%  | 0.14%           | 2.76%       | 11.1% |
| MMB$^{CN}$ | 58.8% | 0.16%       | 2.92%       | 10.7% |

Table 7: Average attention probability (%) allocation of our LayoutLM$^{v3}$+DINO$^{v2}$+CLIP triple-expert model's output on each input.

| Model | Param | VQA$^{v2}$ | GQA | SQA$^{I}$ | VQA$^{T}$ | POPE | MMB | MMB$^{CN}$ | SEED$^{I}$ | MM$^{Vet}$ | **Avg** |
|-------|-------|------------|-----|-----------|-----------|------|-----|-----------|-----------|-----------|---------|
| BLIP-2 (2023b) | 14.1B | 41.0 | 41.0 | 61.0 | 42.5 | 85.3 | – | – | 46.4 | 22.4 | |
| InstructBLIP (2023a) | 8.2B | – | 49.2 | 60.5 | 50.1 | – | 36.0 | 23.7 | 53.4 | 26.2 | |
| VisualGLM (2022) | 8.0B | – | – | – | – | – | 37.6$^{†}$ | 35.5$^{†}$ | 47.0$^{‡}$ | 14.8$^{‡}$ | |
| Shikra (2023b) | 7.3B | 77.4 | – | – | – | – | 58.8 | – | – | – | |
| PandaGPT-13B (2023) | 13B | – | – | – | – | – | 45.4$^{†}$ | 32.0$^{†}$ | 47.6$^{‡}$ | 19.6$^{‡}$ | |
| Qwen-VL-Chat (2023) | 9.6B | 78.2 | 57.5 | 68.2 | **61.5** | – | 60.6 | 56.7 | 58.2 | – | |
| mPLUG-Owl2 (2023) | 8.2B | **79.4** | 56.1 | 68.7 | 54.3 | – | 66.5$^{†}$ | 59.5$^{†}$ | 64.5$^{‡}$ | 35.7$^{‡}$ | |
| Monkey (2024) | 9.8B | – | – | 69.4 | – | – | 59.6$^{†}$ | 54.7$^{†}$ | 64.3$^{‡}$ | **38.1$^{‡}$** | |
| CLIP / LLaVA-1.5 (2023a) | 7.3B | 78.5 | 62.0 | 66.8 | 58.2 | 85.9 | 63.0 | 57.4 | 66.2 | 30.5 | 63.2 |
| **Ours**$_{LayoutLM^{v3}+ConvNeXt+CLIP}$ | 7.9B | 78.5 | 62.9 | 69.8 | 57.6 | **86.6** | 66.6 | 59.8 | 66.2 | 32.5 | 64.5 |
| **Ours**$_{LayoutLM^{v3}+DINO^{v2}+CLIP}$ | 8.8B | 79.1 | **63.6** | 69.0 | 58.4 | 86.5 | **67.4** | **60.0** | **67.5** | 33.6 | **65.0** |

Table 8: The effect of default data on nine benchmarks. **Param.** indicates the number of parameters. The †/‡ marks denote the results from MMB/OpenCompass official, respectively.

## 4 Related Work

**Vision-Language Models (VLMs)** integrate linguistic and visual processing, showing promising results in various applications. VisualGPT (Chen et al., 2022) provided foundational work in image captioning, BLIP series (Li et al., 2022; 2023b) extended capabilities to include visual question answering. Flamingo (Alayrac et al., 2022) and Kosmos-1 (Huang et al., 2023) demonstrated effective multi-modal understanding. LLaVA (Liu et al., 2023b) and MiniGPT-4 (Zhu et al., 2023) utilize projection for connecting vision and language. CogVLM (Wang et al., 2023b) replicated close to double the parameters to build visual experts specializing in visual tokens, while similar to our exploration of positional encoding, they used share-by-one rather than the original approach. Qwen-VL and BLIP series (Bai et al., 2023; Dai et al., 2023b) use the Q-Former network to bridge text and image.

**Multi-Modal Large Language Models (MLLMs)** have been evolving rapidly, with models like ImageBind-LLM (Han et al., 2023) and PandaGPT (Su et al., 2023) incorporating richer modality inputs, including audio and video. There is also a growing focus on region-level parsing (Chen et al., 2023b), text-to-image generation (Wen et al., 2023), and 3D understanding (Xu et al., 2023). These models show that MLLMs can achieve meaningful performance across a range of tasks.

## 5 Conclusion

In this paper, we push the boundaries of VLMs by proposing a novel polyvisual system that closely mirrors the complex and multi-dimensional nature of biological visual processing. Leveraging the unique attributes of diverse visual encoders, our system unifies their strengths to enrich the multimodal understanding of VLMs. Furthermore, we address the challenge of efficiently integrating visual information into language models by introducing techniques such as multi-patch-single-token projection and optimizing positional embeddings. This not only allows us to manage the overflow of vision tokens that typically burdens VLMs but also retains the models' semantic and spatial reasoning capabilities. Through rigorous experiments across a suite of benchmarks, we demonstrate that our polyvisual approach significantly enhances the VLMs' performance, outpacing existing models in accuracy and depth of understanding. These results support our hypothesis that a well-integrated assembly of expert encoders can lead to a substantial improvement in handling complex multimodal inputs.

## 6 Acknowledgements

The authors wish to thank the anonymous reviewers for their helpful comments. This work was partially funded by National Natural Science Foundation of China (No.62441602, 62206057, 62076069), Shanghai Rising-Star Program (23QA1400200), Natural Science Foundation of Shanghai (23ZR1403500), Program of Shanghai Academic Research Leader under grant 22XD1401100.

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

# A  Vision Experts

**CLIP**  learns the image-text alignment through contrastive learning. It is pre-trained on a large-scale dataset consisting of 400M noisy image-text pairs sourced from the internet. The vision encoder of CLIP is a Vision Transformer (ViT) with 300M parameters. The input resolution is fixed to 336×336, and the feature dimension is 1024.[3]

**DINO$^{v2}$**  trains a student network to mimic the behavior of a more powerful teacher network, without the need for any training labels. Two objective functions are utilized for self-supervised pretraining: an image-level object that constrains the CLS tokens from the student network and teacher network, and a patch-level object that is applied to the extracted representations of masked input. The DINO$^{v2}$ vision encoder is a Vision Transformer (ViT) with 1.1B parameters. The input image is preprocessed to 224×224 resolution and the hidden dimension is 1536[4].

**LayoutLM$^{v3}$**  pre-trains multimodal Transformers for Document AI with unified text and image masking. The simple unified architecture and training objectives make LayoutLM$^{v3}$ a general-purpose model for both text-centric and image-centric Document AI tasks. The LayoutLM$^{v3}$ vision encoder is a ViT architecture with 368M parameters. The input image is first preprocessed to the resolution of 224×224 and then encoded to 1024-dimension patch embeddings.[5]

**ConvNeXt**  is a purely convolutional network (ConvNet) that introduces a fully convolutional masked autoencoder framework (FCMAE) and a new global response normalization (GRN) layer to ConvNeXt. ConvNeXt underwent pretraining on the ImageNet-22K dataset, significantly enhancing the performance of the pure ConvNet across various recognition benchmarks. The ConvNeXt vision encoder we used has 200M parameters. The input resolution is 384×384 and the feature dimension is 768.[6]

**SAM**  is trained on a large-scale segmentation dataset, comprising 11 million images and over 1 billion masks, and achieves impressive zero-shot generalization. It is designed to efficiently predict object masks from images with different types of prompts, e.g., text or point. SAM also adopts ViT as a vision encoder with 637M parameters. The input resolution and hidden dimension are both larger, i.e., 1024×1024 and 1280, respectively.[7]

**MAE**  aims to reconstruct the original image given only partial observations (25% of the patches). The ViT-Huge encoder paired with MAE achieved a new record at the time on the ImageNet-1K dataset with an accuracy of 87.8% and generalized very well. The MAE vision encoder has 630M parameters, while input resolution and hidden dimension are 1024×1024 and 1280.[8]

---

[3]https://huggingface.co/openai/clip-vit-large-patch14-336
[4]https://huggingface.co/facebook/dinov2-giant
[5]https://huggingface.co/microsoft/layoutlmv3-large
[6]https://huggingface.co/laion/CLIP-convnext_large_d_320.laion2B-s29B-b131K-ft-soup
[7]https://huggingface.co/facebook/sam-vit-huge
[8]https://huggingface.co/facebook/vit-mae-huge

## B  Analysis

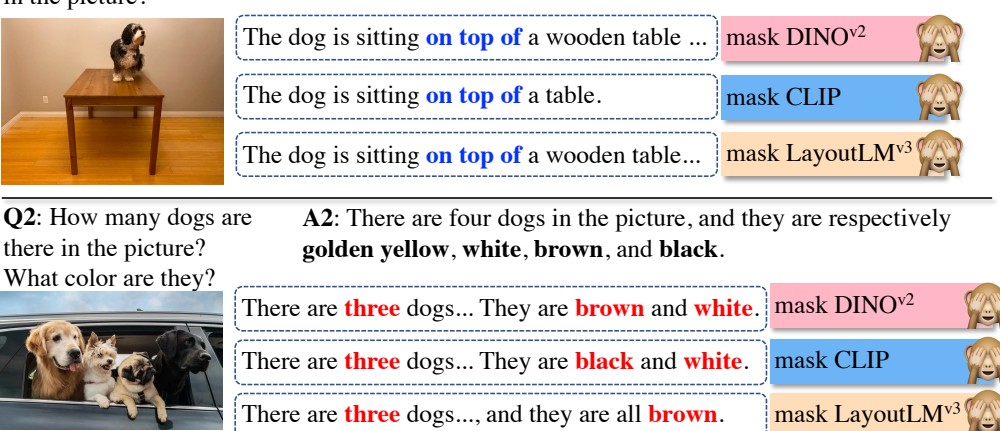

Figure 4: The perturbation experiments on the triple-expert LayoutLM$^{v3}$+DINO$^{v2}$+CLIP model, the specific perturbation is to mask all output of the corresponding vision expert.

Figure 4 presents perturbation experiments conducted on the same triple-expert. The outputs of each expert were individually masked to assess their influence on the model's response. In Case 1, our model is queried with a straightforward question: "Where is the dog in the picture?" The experiment demonstrates that, irrespective of which visual expert is masked, the two unmasked channels sufficiently identify the location as "on top of". However, the presence of CLIP experts enriches the details by specifying "wooden table" over a generic "table". Case 2 poses a more complex question to our model: "How many dogs are there in the picture? What colors are they?" Here, the perturbation tests reveal that the collective operation of all three experts is necessary for an accurate response. The omission of any single expert leads to erroneous answers, underlining the diverse informational scope captured by each visual channel within the poly-visual-expert VLM. This underscores the importance of channel integration in multimodal tasks, as certain nuances can be missed when relying on a solitary channel or single-expert VLM.

## C   Datasets & Evaluation & Hyperparameters

**Datasets**   During the pre-training stage, we utilized the LCS-558K dataset[9], which comprises ~558K image-text pairs from the LAION-CC-SBU, annotated with BLIP-generated captions. During the SFT stage, We use the default SFT data from LLaVA-1.5, which is ~665k[10], containing VQA, OCR, region-level VQA, visual conversation, and language conversation data.

**Evaluation**   Including $VQA^{v2}$ (2017); GQA (2019); $SQA^{I}$ : ScienceQA-IMG (2022); $VQA^{T}$: TextVQA (2019); POPE (2023c); MMB & $MMB^{CN}$: MMBench & MMBench-Chinese *dev* results (2023c); $SEED^{I}$ : SEED-Bench-IMG (2023a); MM-Vet (2023).

**Hyperparameters**   For main results, we keep all training hyperparameters roughly the same as the LLaVA series (Liu et al., 2023b;a). We present a detailed description of the hyperparameters in Table 9. For the MLP fusion network, we set $m$ in $m$-patches-one-token from 1 to 16 to avoid exceeding the maximum length for training and inference. For the Q-Former fusion network, we set the number of queries per expert to match the number of outputs from the MLP fusion network. The parameters of the Q-Former fusion network are initialized using the pre-training parameters of BLIP-2 (Li et al., 2023b).

| Hyperparameter | Pretrain | Finetune |
|---|---|---|
| batch size | 256 | 128 |
| lr | 1e-3 | 2e-5 |
| lr schedule | cosine decay | |
| lr warmup ratio | 0.03 | |
| weight decay | 0 | |
| epoch | 1 | |
| optimizer | AdamW | |
| DeepSpeed stage | 2 | 3 |

Table 9: Hyperparameters of our model's pretrain and finetune.

---

[9]https://huggingface.co/datasets/liuhaotian/LLaVA-Pretrain
[10]https://huggingface.co/datasets/liuhaotian/LLaVA-Instruct-150K

# D   Case Study

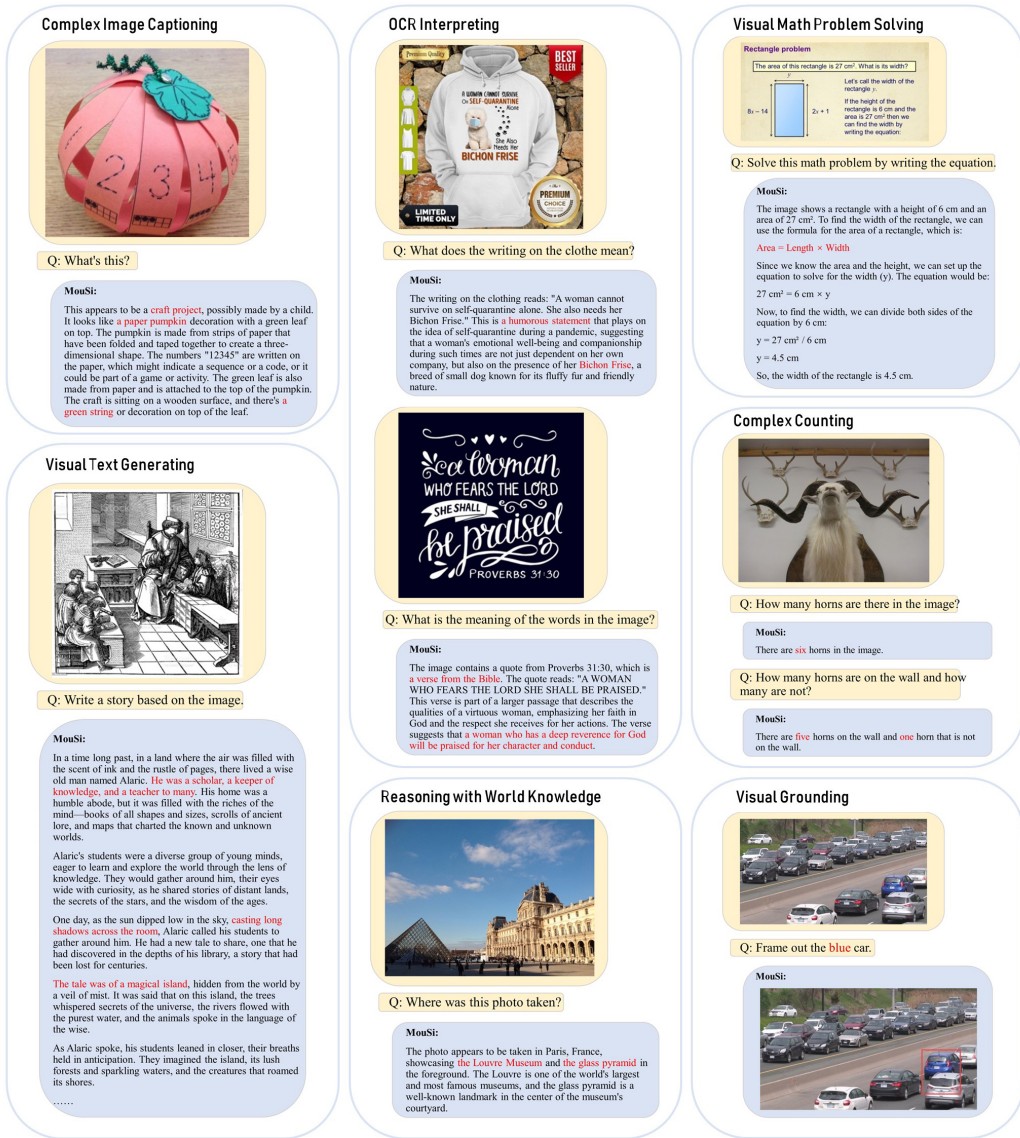

Figure 5: Qualitative examples generated by our poly-vision-expert model.

Figure 5 shows the case study of our model on seven tasks, including Complex Image Captioning, Visual Text Generating, OCR Interpreting Reasoning with World Knowledge, Visual Math Problem Solving, Complex Counting, and Visual Grounding. Our model is able to successfully follow a variety of multimodal instructions, allowing for flexible interaction with humans.

