# OpenReview forum: "Poly-Visual-Expert Vision-Language Models"
_colmweb.org/COLM/2024/Conference — COLM_

### Official Review · Reviewer_oxQg · 2024-05-07

**Rating:** 7
**Confidence:** 4
**Ethics Flag:** 1

**Summary:**

This paper proposes a way to fuse together multiple vision encoders within a vision-language model setup. They show that a) using two and three vision encoders (that are pre-trained on complementary tasks and datasets like image-text contrastive learning, image segmentation, and OCR) can improve upon using a single vision encoder (i.e. CLIP), b) a simple MLP does better than a more complicated transformer-based network (Q-former) in fusing the image and text modalities from the image encoder and the LLM, and c) improving the efficiency of VLMs with respect to input tokens by eliminating a lot of redundancy in the positional embedding space for images.

**Reasons To Accept:**

* I believe this might be the first VLM that combines multiple vision encoders (trained on different objectives and tasks), and they do show by and large improvement on a wide variety of tasks
* evaluation in multiple languages

**Reasons To Reject:**

* would have been great to see further analysis on Table 2 and Table 8. For example in Table 2, why is VQA anomalous? In Table 8, we see results are a lot more mixed with 3 experts. Is this a limitation of the approach that could be highlighted in the main text?
* it doesn't seem like, compared to "origin", the parameter sharing schemes in 3.2.2 have much effect on anything? The differences with the baseline are small, in fact smaller than the effect of expert order (Table 9)

---

> ### Author Rebuttal · Authors · 2024-05-29
>
> Thank you very much for your providing insightful suggestions. We have addressed each of the issues you raised. We will make the revisions to our manuscript.
>
> > **Response to Q1: would have been great to see further analysis on Table 2 and Table 8. For example in Table 2, why is VQA anomalous? In Table 8, we see results are a lot more mixed with 3 experts. Is this a limitation of the approach that could be highlighted in the main text?**
> - Thank you for pointing out some anomalies that require further analysis. Using the TextVQA results in Table 2 as an example, we found that it differs from other benchmarks as it emphasizes the model's ability to read and reason about text information in images. When the CLIP expert collaborated with the LayoutLM expert, the performance on TextVQA improved by 0.7, benefiting from LayoutLM's document-level OCR pre-training. Conversely, combining with other experts decreased performance (ranging from -0.3 to -1.3) because their pre-training processes did not emphasize text information within images.
> - This *negative transfer effect* indicates that, in a multi-expert system, the specialties and capabilities of different experts may lead to conflicts or contradictions. It is a limitation as you mentioned of our current approach, suggesting that expert combinations should be flexible and changeable when handling different tasks. This will be a focus of our future work, potentially reflected in MouSi-2.
>
> > **Response to Q2: it doesn't seem like, compared to "origin", the parameter sharing schemes in 3.2.2 have much effect on anything? The differences with the baseline are small, in fact smaller than the effect of expert order (Table 9)**
> - There may be a misunderstanding here, so we'd like to clarify that the **Different Position Schemes** we propose aim to resolve the out-of-max-length issue in VLMs, particularly when handling multiple vision experts (e.g., triple-experts) or input images (e.g., video). The aim is not to improve the performance of downstream tasks. As a result, the "Share-all" method consumed the fewest positional encodings (one position per image) while maintaining performance.

---

> > ### Comment · Reviewer_oxQg · 2024-06-03
> > **Rebuttal acknowledgement**
> >
> > I have acknowledged the response and have kept my score as is.

---

### Official Review · Reviewer_RrkK · 2024-05-11

**Rating:** 6
**Confidence:** 4
**Ethics Flag:** 1

**Summary:**

This paper investigates how to ensemble multiple image encoders to provide a visual input to an LLM. The authors explore the use of six different visual encoders (eg. CLIP, DINO, MAE) and find that, often, using three encoders is better than using two encoders, which in turn is better than the best single encoder. To combine the visual tokens from multiple visual encoders, the authors devise a fusion network that reduces the number of tokens per visual encoder while mapping the image embedding to the LLM’s input dimension. Experiments on 9 datasets show that the proposed approach improves upon using a single image encoder as well as on related baselines.

**Questions To Authors:**

1. In Section 2.1, you state that “the image is processed in _sequence_ through the [visual] encodings”; do you mean in parallel (looking at Figure 2)?
2. In Table 2, could you please add the absolute performance of a CLIP-based model in the first row for better comparability?
3. The model size increases by 1.5B parameters when using three encoders. How does this influence inference time/memory in practice? This is a key information (along with accuracy) to decide whether/when to use this method.

**Reasons To Accept:**

1. The authors study how to effectively and efficiently incorporate multiple image encoders with different strengths to improve visual understanding for multimodal tasks
2. The proposed approach is simple yet often yields better results than using a single image encoder, especially on SQA and MMB

**Reasons To Reject:**

1. Albeit being generally well written, the paper is not easy to follow. Several key results are delegated to the appendix, without much discussion in the main paper. For example, it is said that triple-encoders outperform other methods (ie. are the strongest proposed model) but are however not discussed in the main paper. Another example is at the beginning of Section 2.2, where you said that you chose six image encoders “after extensive investigation”; which is not shared with the readers. Lastly, you add a whole Section 3.4 on “data enhancement” without ever saying what you do in the main body, letting the reader wonder what this might be about.
2. The performance gains are within 1 percentage point for most datasets. We can observe larger gains on SQA and MMB, but the overall boost between the three-expert model and the CLIP-only model is less than 1 percentage point in the final “Data enhancement” results.
3. Moreover, the results include evaluation on both MMB and MMB-CN (a version of MMB translated into Chinese). If the goal is to assess the ability of the model to perform multiple tasks, having MMB-CN is redundant as it assesses the same ability as MMB. If the goal is to evaluate multilingual capabilities, there are far better benchmarks to do so (eg. MaRVL, Liu+EMNLP’21; IGLUE, Bugliarello+ICML’22; XM3600, Thapliyal+EMNLP’22).
4. One of your main contributions was to tackle the “challenge of visual token overflow”. I was not aware of this challenge, was it introduced by someone else before (if so, please cite) or was it something you first encountered in this study? In the latter case, I would rather see it as an ablation of positional encoding more than a challenge itself, as one can simply not add positional encodings (again) after the visual embeddings, since these normally include positional encodings in their respective encoders (which was your final recommendation too)
5. The related work would benefit from discussing other approaches for ensembling (both for VLMs and not) as well as for token merging/dropping.

---

> ### Author Rebuttal · Authors · 2024-05-30
>
> Thank you for your insightful comments.
>
> ---
>
> > **Response to W1&W5: Several key results are delegated to the appendix & More discussion on related work**
> - We apologize for moving key results to the appendix due to page limits. Fortunately, noticed by the PCs' email that there will be an extra page. We promise to add the necessary results and descriptions to enhance readability. Thanks for your helpful suggestions.
>
> ---
>
> > **Response to W2: Most performance gains less than 1**
> - Our performance gains are comparable to (or even higher than) other recently published works, e.g., compared to LLaVA-1.5 on 6/8 shared datasets, the average gains of Monkey and mPLUG-Owl2 are +0.0 (59.6) and +0.7 (60.3). While our default and data enhanced models are +2.3 (61.9) and +0.5 (63.8).
> ---
>
> > **Response to W3: Evaluation on MMB-CN**
> - Since SFT data contains both EN and CN, we evaluated on MMB-CN (in line with LLaVA).
>
> ---
>
> > **Response to W4: Challenge of visual token overflow**
> - Since LVLMs apply LLM's position encoding(PE) again to vision tokens(i.e., double PE), they are prone to encounter out-of-max-length issues, especially when handling multiple visual experts or multiple input images. Our experiments confirmed that vision tokens don’t degrade baseline performance even without second PE. We effectively mitigated this issue by proposing multi-patch-single-token projection and Share-all PE.
>
> ---
>
> > **Response to Q1: Do you mean in parallel**
> - Yes, multiple experts encode images independently and in parallel. We will revise it.
>
> ---
>
> > **Response to Q2: Absolute performance of CLIP**
> - We will add it.
>
> ---
>
> > **Response to Q3: Inference time, Memory**
>
>  |Model|Param|Memory(G)|Inference time(s)/Example|
>  |-|-|-|-|
>  |1-expert(0.3B-CLIP)|7.3B|15.9 1.00x|0.120 1.00x|
>  |1-expert(1.9B-CLIP)|8.9B|19.3 1.21x|0.177 1.48x|
>  |2-expert-ours|8.4B|16.9 1.06x|0.145 1.21x|
>  |3-expert-ours|8.8B|18.2 1.14x|0.149 1.24x|
>
> - We conducted inference time and memory usage experiments with MMB. For fair comparison, we added a single encoder(1.9B-CLIP) baseline with a similar number of total parameters.
>
> - Compared to LLaVA, our selected(from paper) 2/3 experts' inference time increased by 1.21x/1.24x, memory increased by 1.06x/1.14x. The additional inference cost is not significant.
>
> - Our inference cost is even lower than that of a single encoder with the same parameter size, demonstrating the efficiency of the m-patch-one-token method and parallel encoding with multi-visual-experts.

---

> > ### Author Response · Authors · 2024-06-04
> > **Looking Forward to Further Discussion**
> >
> > Response to the Reviewer RrkK
> >
> > We sincerely appreciate the detailed feedback you provided on our work.
> > We have addressed each of your concerns with **additional inference time and memory usage experiments** and clarification of the contribution of our work. The details in our response are summarized as follows:
> >
> > - **We performed additional inference time and memory usage experiments**. Our results show that: **our inference cost is even lower than that of a single encoder with the same parameter size, demonstrating the efficiency of the m-patch-one-token method and parallel encoding with multi-visual-experts.**
> >
> > - We elaborated that our performance gains are **comparable to (or even higher than)** other recently published works.
> >
> > - Our experiments confirmed that vision tokens don’t degrade baseline performance even without second PE. We effectively mitigated this issue by proposing multi-patch-single-token projection and Share-all PE.
> >
> > - **We answered all your questions and addressed your concerns**, and we promise to add the necessary results and descriptions to enhance readability.
> >
> > As we approach the end of the rebuttal process, we eagerly await your response to ensure our rebuttal has adequately addressed your concerns.  Should you have any more questions, we are ready to respond before the end of the rebuttal period.  We appreciate your time and consideration.

---

> > > ### Comment · Reviewer_RrkK · 2024-06-06
> > > **Acknowledgement of Rebuttal**
> > >
> > > Dear authors,
> > >
> > > Thank you for replying to my questions. I have also read the other reviews and responses.
> > > In light of them, I have increased my score.
> > >
> > > It would however be much better if you (1) did not average results on MMB-CN along with other English tasks, and (2) evaluated on other multilingual benchmarks (even if just on the CN split) as other benchmarks have been created for this purpose. In this aspect, just because one paper (LLaVA) had suboptimal experimental settings, you should not blindly follow them.

---

> > > > ### Author Response · Authors · 2024-06-06
> > > > **Response to Reviewer**
> > > >
> > > > Dear Reviewer RrkK,
> > > >
> > > > Thank you again for your valuable comments and constructive suggestions. We will follow your suggestions: (1) to evaluate English and Chinese tasks separately instead of mixing them up, and (2) to supplement more Chinese evaluation sets.

---

### Official Review · Reviewer_BvJn · 2024-05-12

**Rating:** 8
**Confidence:** 3
**Ethics Flag:** 1

**Summary:**

In this paper, authors present the use of ensemble experts technique to synergize the capabilities of individual visual encoders, including those skilled in image-text matching, OCR, image segmentation, etc. The proposed approach MouSi have a poly-expert fusion network, a pre-trained open-source LLM and a multi-expert visual encoder. Experiments performed with various benchmarks showcase combing multiple experts provide better performance in contrast to single expert.

**Questions To Authors:**

Comments to Authors

1. Is there an effective way to combine the experts or all known experts for a task is used and the model will figure it out which one to use? Table-2 showed that double vision experts are useful, but these combinations can become overwhelming when there are large number of experts available for a task.

2. Although, Section-3.5 mention different case studies by MouSi. Is MouSi capable of comprehending different languages text also?

3. Although multiple experts boos performance, how effective it is for the time-sensitive applications which require fast inference, still single expert is the approach for them? Some discussion on the trade-off will be useful. Also during inference, even though parameter space has increased, will all parameters will be used?

Minor Comments

1. Table-2 figure is unreadable without zoom. Please make it more visible.

**Reasons To Accept:**

1. Proposed method compresses the local visual information by multi-patch-one-token for better transmission efficiency and reduces the quadratic computational cost of subsequent processing of VLMs.

2. Explores different positional encoding schemes to effectively address the issue of position encoding waste. To be specific, sharing one position for all patches and 2D positional encoding.

3. Experiments are showcased with different combination of experts (i.e., VLMs)

**Reasons To Reject:**

1. Mixture-of-experts idea has been explored in the literature, and the proposed approach MouSe is built on the same principle

2. MLP fusion network or Q-Former network has been a standard approach used by VisualLLMs. The proposed Poly-Fusion Network is very similar in architecture.

3. Experimental study combining different available models and approaches. No new novel method proposed.

---

> ### Author Rebuttal · Authors · 2024-05-30
>
> Thanks for your comments and helpful feedback.
>
> ---
>
> > **Response to Q1 & Q3: Mixture-of-experts idea has been explored in the literature & no new novel method proposed**
> - We agree and will supplement the Related Work section with MoE-related literature. We'd like to emphasize that, to the best of our knowledge (also mentioned by Reviewer oxQg), we are the first to apply this idea to LVLMs.
>
> ---
>
> > **Response to Q2: Poly-Fusion network is very similar to the standard MLP or Q-Former network**
> - We agree and would like to clarify that the primary goal of the Poly-Fusion network is not to innovate the standard MLP or Q-Former networks, but rather 1) to demonstrate how MLP or Q-Former networks can be adapted from a single vision expert to multiple vision experts, and 2) to determine which network, MLP or Q-Former, is more suitable for multiple experts (the conclusion is that MLP is significantly better).
>
> ---
>
> > **Response to Q4: Is there an effective way to combine the experts for a task and the model will figure out which one to use?**
> -  MouSi has demonstrated the effectiveness of Poly-Visual-Expert, but we have also observed negative transfer effects (potential conflicts between experts). This suggests that expert combinations should be flexible and changeable when handling different tasks. A feasible solution is to add a trainable routing module of experts, allowing Poly-Visual-Fusion networks to *sparsely activate* corresponding experts for specific tasks. This will be a focus of our future work, potentially reflected in MouSi-2.
>
> ---
>
> > **Response to Q5: Is MouSi capable of comprehending different languages text also?**
> - MouSi's pre-training dataset is only in English, while the SFT dataset includes both Chinese and English. Since the LLM module (Vicuna-1.5) has multilingual capabilities, we found that MouSi can generalize to some other languages (such as French or German), but it is not always successful.
>
> ---
>
> > **Response to Q6: How effective it is? Also during inference, will all parameters be used?**
>
>  |Model|Param|Memory(G)|Inference time(s)/Example|
>  |-|-|-|-|
>  |1-expert(0.3B-CLIP)|7.3B|15.9 / 1.00x|0.120 / 1.00x|
>  |1-expert(1.9B-CLIP)|8.9B|19.3 / 1.21x|0.177 / 1.48x|
>  |2-expert-ours|8.4B|16.9 / 1.06x|0.145 / 1.21x|
>  |3-expert-ours|8.8B|18.2 / 1.14x|0.149 / 1.24x|
>
> - The increased inference time and memory on three experts are *not significant*, benefiting from the proposed Poly-Fusion network.
> - During inference, all parameters are used.

---

> > ### Author Response · Authors · 2024-06-04
> > **Looking Forward to Further Discussion**
> >
> > Response to the Reviewer BvJn
> >
> > We sincerely appreciate the detailed feedback you provided on our work.
> > We have addressed each of your concerns with **additional inference time and memory usage experiments** and clarification of the contribution of our work:
> >
> > - **We performed additional inference time and memory usage experiments**. Our results show that: **our inference cost is even lower than that of a single encoder with the same parameter size, demonstrating the efficiency of the m-patch-one-token method and parallel encoding with multi-visual-experts.**
> >
> > - Our primary goal of the Poly-Fusion network is not to innovate the standard MLP or Q-Former networks, but rather 1) to demonstrate how MLP or Q-Former networks can be adapted from a single vision expert to multiple vision experts, and 2) to determine which network, MLP or Q-Former, is more suitable for multiple experts.
> >
> > - MouSi has demonstrated the effectiveness of Poly-Visual-Expert, we will add a trainable routing module of experts, allowing Poly-Visual-Fusion networks to sparsely activate corresponding experts for specific tasks, potentially reflected in MouSi-2.
> >
> > - MouSi can generalize to some other languages (such as French or German).
> >
> > - We agree and will supplement the Related Work section with MoE-related literature. We'd like to emphasize that, to the best of our knowledge (also mentioned by Reviewer oxQg), we are the first to apply this idea to LVLMs.
> >
> >
> > As the rebuttal process draws to a close, we eagerly await your response to ensure that our rebuttal adequately addresses your concerns. If you have any further questions, we stand ready to respond before the end of the rebuttal period. Thank you for your time and consideration.

---

> > > ### Comment · Reviewer_BvJn · 2024-06-06
> > > **Acknowledgement of Rebuttal**
> > >
> > > Thanks for the details about the concerns raised in the review. It address the concerns. I have acknowledged the response and have kept my score as is.

---

### Official Review · Reviewer_a4mu · 2024-05-16

**Rating:** 7
**Confidence:** 4
**Ethics Flag:** 1

**Summary:**

Current VLM only uses a single visual encoder to obtain image information, which may not capture all the useful hints in the image. This paper proposes an ensemble framework that uses more than one visual encoder to get image information and introduces a fusion network to unify the processing of outputs from different visual encoders. Experiments on downstream tasks show that incorporating more than one visual encoder could significantly increase the model's performance in different aspects.

**Questions To Authors:**

1. The ensemble model contains more parameters than baselines, it might be unfair for comparison.

2. [minor] It would be better if the strengths and weaknesses of different visual experts could be summarized more clearly, rather than just using case studies to demonstrate them.

**Reasons To Accept:**

1. This paper proposes a simple but effective method to take advantage of different pre-trained visual encoders. Experiments show the proposed method could significantly outperform the method with only a single visual encoder.

2. The improvements of the ensemble method further illustrate that the information obtained by different visual encoders is different. Future work might consider how to help one visual expert encode more details.

3. The ablation study is comprehensive and is very helpful in understanding the contribution of different components.

**Reasons To Reject:**

NA

---

> ### Author Rebuttal · Authors · 2024-05-30
>
> Thank you for your encouraging comments and helpful feedback. We have addressed each of the issues you raised. We will make the revisions to our manuscript.
>
> ---
>
> > **Response to Q1: The ensemble model contains more parameters than baselines, it might be unfair for comparison.**
>
> - Thank you for your comments. First, we would like to emphasize that the ensemble model has the *same* number of *learnable parameters* as the baseline because the visual encoders, whether there is one or multiple, are always frozen (as in the case with most LVLMs).
>
> - Second, to make the total parameters of triple-experts and single-experts comparable, we did our best to find a 1.9B CLIP encoder (CLIP-ViT-bigG-14-laion2B-39B-b160k), and trained a new single-expert baseline (8.9B). The results show that our method remains state-of-the-art with a fair comparison. Additionally, using a 1.9B CLIP expert even hurt the baseline's performance, demonstrating that the capability of visual experts is not related to the number of their parameters (as also evidenced by Table 7 in Appendix C of our paper).
>
>   |Model|Param.|VQAv2|GQA|VQA-T|POPE|MMB|MMB-CN|SEED-I|MM-Vet|Avg@8|
>   |-----|------|----------------|---|--------------|----|---|--------------|-------------|------|-----|
>   |Single-expert(0.3B CLIP-Res-336)|7.3B|78.5|62.0|58.2|85.9|63.0|57.4|66.2|30.5|62.7|
>   |Single-expert(1.9B CLIP-Res-224)|8.9B|73.1|52.8|50.4|82.2|64.7|58.6|59.0|27.9|58.6|
>   |Triple-expert|8.8B|79.1|63.6|58.4|86.5|67.4|60.0|67.5|33.6|**64.5**|
>
> ---
>
> > **Response to Q2: [minor] It would be better if the strengths and weaknesses of different visual experts could be summarized more clearly, rather than just using case studies to demonstrate them.**
>
> - Inspired by your and reviewer oxQg's insightful comments, we believe that an in-depth analysis of the experimental results can yield some valuable conclusions.
>
> - Using the TextVQA results in Table 2 as an example, we found that it differs from other benchmarks as it emphasizes the model's ability to read and reason about text information in images. When the CLIP expert collaborated with the **LayoutLM** expert, the performance on TextVQA improved by 0.7, benefiting from LayoutLM's document-level OCR pre-training. Conversely, combining with other experts decreased performance (ranging from -0.3 to -1.3) because their pre-training processes did not emphasize text information within images. This clearly demonstrates **LayoutLM**'s strength in text OCR tasks.

---

> > ### Author Response · Authors · 2024-06-04
> > **Looking Forward to Further Discussion**
> >
> > Response to the Reviewer a4mu
> >
> > We sincerely thank you for the thoughtful and constructive feedback.
> > We have addressed each of your concerns with **additional experiments for fair comparison** and we analyzed the experimental results in more depth to yield some valuable conclusions.
> >
> > As the rebuttal process draws to a close, we eagerly await your response to ensure that our rebuttal adequately addresses your concerns. If you have any further questions, we stand ready to respond before the end of the rebuttal period. Thank you for your time and consideration.

---

### Decision · Program_Chairs · 2024-07-10

**Decision:**

Accept

**Comment:**

This paper proposes a vision--language model that combines multiple expert vision encoders---similarly to MoE work in the LM space. The authors also proposes a method to reduce the number of positional embeddings used by various models. They show the effectiveness of their approach across a range of benchmarks and perform ablations to examine the effectiveness of different component of their model. This paper contributes to the MoE work in the multimodal space, and the future work can build on it. However, I encourage the authors to address the reviewers' feedback by (1) adding discussion around the related work (2) including analysis of inference time, memory, and the effect of number of parameters (3) discussion around strengths/weaknesses of each expert.